# A VSV-based assay quantifies coronavirus Mpro/3CLpro/Nsp5 main protease activity and chemical inhibition

Emmanuel Heilmann [1✉], Francesco Costacurta[1,6], Stephan Geley [2,6], Seyad Arad Mogadashi[3,6], Andre Volland[1,6], Bernhard Rupp[4], Reuben Stewart Harris [3,5] & Dorothee von Laer [1✉]

Protease inhibitors are among the most powerful antiviral drugs. However, for SARS-CoV-2 only a small number of protease inhibitors have been identified thus far and there is still a great need for assays that efficiently report protease activity and inhibition in living cells. Here, we engineer a safe VSV-based system to report both gain- and loss-of-function of coronavirus main protease (M$^{pro}$/3CLpro/Nsp5) activity in living cells. We use SARS-CoV-2 3CLpro in this system to confirm susceptibility to known inhibitors (boceprevir, GC376, PF-00835231, and PF-07321332/nirmatrelvir) and reevaluate other reported inhibitors (baicalein, ebselen, carmofur, ethacridine, ivermectin, masitinib, darunavir, and atazanavir). Moreover, we show that the system can be adapted to report both the function and the chemical inhibition of proteases from different coronavirus species as well as from distantly related viruses. Together with the fact that live cell assays also reflect compound permeability and toxicity, we anticipate that this system will be useful for both identification and optimization of additional coronavirus protease inhibitors.

[1] Institute of Virology, Medical University of Innsbruck, Innsbruck, Austria. [2] Division of Pathophysiology, Medical University of Innsbruck, Innsbruck, Austria. [3] Department of Biochemistry, Molecular Biology and Biophysics, Masonic Cancer Center, Institute for Molecular Virology, Center for Genome Engineering, University of Minnesota, Minneapolis, MN 55455, USA. [4] Institute of Genetic Epidemiology, Medical University of Innsbruck, Innsbruck, Austria. [5] Howard Hughes Medical Institute, University of Minnesota, Minneapolis, MN 55455, USA. [6] These authors contributed equally: Francesco Costacurta, Stephan Geley, Seyad Arad Mogadashi, Andre Volland. ✉email: emmanuel.heilmann@i-med.ac.at; virologie@i-med.ac.at

The transmission of severe acute respiratory syndrome coronavirus 2 (SARS-CoV-2) into the human population in late 2019 is responsible for a worldwide pandemic. To meet the need for effective therapies, approved drugs, and potential inhibitors have been screened for antiviral activity[1–8]. Viral enzymes such as polymerases and proteases are especially promising targets because they are essential for virus replication[9] and both, polymerase and protease inhibitors, have proven to be highly effective against hepatitis C virus (HCV)[10,11] and human immunodeficiency virus (HIV)[12,13]. Indeed, a first protease inhibitor, Paxlovid, has recently been authorized by the U.S. FDA for emergency use in high-risk SARS-CoV-2-infected individuals (EUA 105 Pfizer Paxlovid).

After SARS-CoV-2 infection, two polyprotein chains (p1a, p1ab) containing the non-structural proteins are translated directly from the positively orientated RNA genome[14]. Included in these two polyproteins are two proteases that must first be released by self-cleavage before they can process the other non-structural protein precursors[15,16]. The 3-chymotrypsin-like protease (3CLpro or main protease) of SARS-CoV-2 cleaves 11 sites in the polyproteins, beginning with autocatalytic cis-cleavage of N- and C-terminal recognition sequences and continuing with trans-cleavage of distant recognition sequences between each distinct non-structural protein. The papain-like protease (PLpro) cleaves 3 additional sites[7,17]. Both 3CLpro and PLpro are interesting antiviral targets. A direct approach to test inhibitors of these enzymes would be to study inhibition of SARS-CoV-2. However, handling active virus requires biosafety level 3 facilities[18], which are not available in most institutions. Biochemical fluorescence resonance energy transfer-based assays have been used to safely (biosafety level 1) identify SARS-CoV-2 protease inhibitors, including GC376 and GRL0617[5,6]. However, these and other in vitro assays necessarily require the use of mature enzymes, meaning they cannot readily measure the earliest stage of activation, i.e., auto-processing or cis-cleavage, which is essential for virus infection and pathogenesis.

To facilitate SARS-CoV-2 3CLpro characterization and drug development, we sought to create a safe, biosafety level 1-based cellular assay based on our prior work using vesicular stomatitis virus (VSV)[19]. VSV is a non-segmented negative-strand RNA virus of the family of Rhabdoviridae[20]. VSV has five essential proteins: nucleoprotein (N), phosphoprotein (P), matrix protein (M), glycoprotein (G), and polymerase (L)[21,22]. Three of these viral proteins tolerate intramolecular insertions[23–25] and functional N- and C-terminal tags have yet to be described. We recently leveraged this knowledge by generating two chimeric VSV constructs. In the inhibitor-off system, the viral polymerase L is tagged on the N-terminus with HIV-1 protease, eGFP, and an HIV-1 protease cleavage site[19]. Only after tag removal by proteolytic cleavage is L protein function restored and VSV able to replicate. Thus, protease inhibitors prevent cleavage and inhibit virus replication. In the inhibitor-on system, the HIV-1 protease flanked by its cognate cleavage sites are inserted into the viral P protein[19]. This chimeric construct is unable to replicate due to HIV-1 protease catalyzing the cleavage of the VSV P protein into separate non-functional pieces. However, HIV-1 protease inhibition with existing drugs prevents internal P protein cleavage, restores P protein function (despite the insertion) and virus replication. Both inhibitor-off and inhibitor-on assays are visualized readily through a fluorescence reporter inserted into the VSV genome. Importantly, protease inhibition strength is directly proportional to reporter signal, with weak inhibitors eliciting weak signal and strong inhibitors strong signal. Here, we engineer these inhibitor-off and inhibitor-on systems to report SARS-CoV-2 3CLpro inhibition and test a panel of reported inhibitors. Altogether, these cell-based SARS-CoV-2 3CLpro assays have the potential to screen for novel and highly effective drugs against SARS-CoV-2.

## Results

**An inhibitor-off system for quantifying 3CLpro activity and chemical inhibition**. Based on our previous experience designing conditional viruses with the HIV-1 protease[19], we generated an inhibitor-off VSV-based system with SARS-CoV-2 3CLpro. The inhibitor-off system is based on a polyprotein consisting of an N-terminal reporter, followed by the SARS-CoV-2 3CLpro, and the C-terminal VSV polymerase L. Cognate 3CLpro cleavage sites are located between the reporter and 3CLpro as well as between 3CLpro and L. This virus is replication-proficient due to efficient 3CLpro-catalyzed cleavage and release of functional L. Virus replication leads to an accumulation of strong reporter signal over time (e.g., GFP and luciferase in Supplementary Fig. 1a–c).

To reduce the biosafety requirements of this system from biosafety level 2[26] to biosafety level 1, we replaced the viral L protein with a fluorescent reporter, dsRed[27], and used lentiviral transduction to express the GFP-3CLpro-L protein fusion in trans in 293T cells (Fig. 1a–c). In this system, 3CLpro is expected to undergo maturation by auto-processing or cis-cleavage, which catalyzes the release of an untagged, functional L protein to promote VSV-ΔL-dsRed replication. Thus, the dsRed fluorescent signal resulting from virus replication could be readily quantified using fluorescence microscopy or a high-throughput plate reader such as a FluoroSpot Counter (Fig. 1d, e). Moreover, bona fide 3CLpro inhibitors such as GC376 suppressed the accumulation of dsRed signal in a dose-responsive manner with low concentrations having little effect and high concentrations completely suppressing virus replication (Fig. 1e). We also noticed some cytotoxicity in the uninhibited system, which may be due to cleavage of cellular proteins by 3CLpro[28,29]. Accordingly, protease inhibition also restored cell viability in a dose-responsive manner (Fig. 1e).

**An inhibitor-on system for quantifying 3CLpro activity and chemical inhibition**. To augment the inhibitor-off system described above, we designed an inhibitor-on system, in which the viral P protein is disrupted by 3CLpro and its cognate cleavage sites (Supplementary Fig. 1d, e). This chimeric construct is unable to replicate due to proteolytic disruption of the VSV P protein; however, addition of the 3CLpro inhibitor GC376 prevented P protein cleavage and restored virus replication as evidenced by reporter fluorescence (Supplementary Fig. 1f).

The biosafety of this initial construct was also improved from biosafety level 2 to biosafety level 1 by splitting it into two components (Fig. 2a–c). A lentiviral construct was used to express VSV P:3CLpro in BHK21 cells. Subsequent infection with VSV-ΔP-dsRed[27] partially reconstituted the system, but virus replication was still defective due to 3CLpro-dependent auto-proteolysis of the VSV P protein. Importantly, treatment of these cells with the 3CLpro inhibitor GC376 caused a dose-responsive restoration of P protein function and virus replication, as quantified by the robust accumulation of dsRed-positive cells (Fig. 2d). We also noticed a diminution of fluorescent signal and cell viability at higher GC376 concentrations, which is likely due to compound toxicity.

**Cleavage site preferences of the inhibitor-off and inhibitor-on systems**. SARS-CoV-1 3CLpro first cuts at the N-terminal cleavage site and then undergoes a structural rearrangement to promote cleavage at the C-terminal site[30,31]. The N- and C-terminal cleavage events are therefore distinct. The inhibitor-on system described here provides a unique opportunity to ask whether

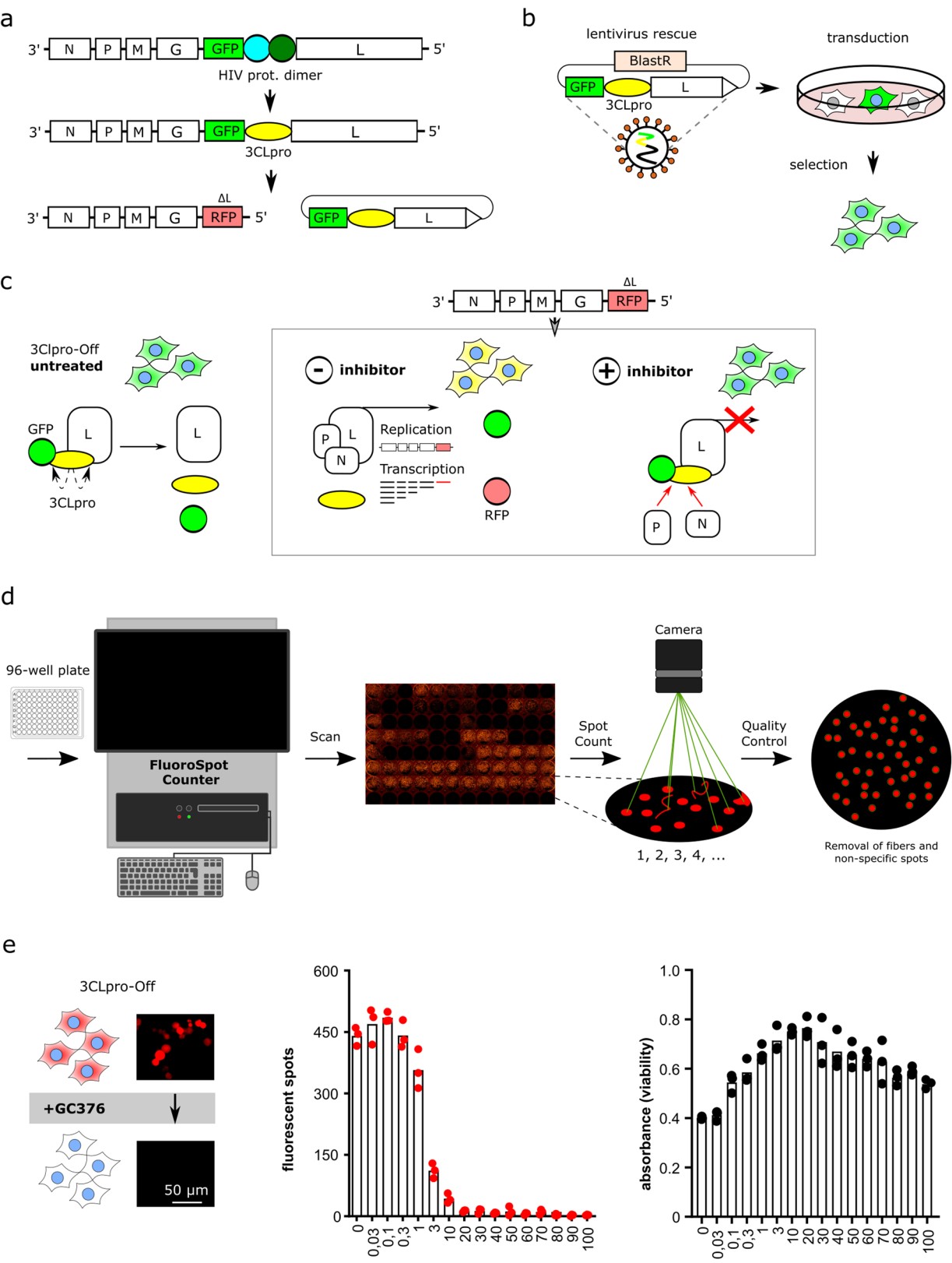

**Fig. 1 Design principles of the inhibitor-off system for quantifying 3CLpro activity and chemical inhibition. a–c** Schematics of the constructs leading to the current biosafety level 1 system consisting of VSV-ΔL-dsRed and Lenti-GFP-3CLpro-L with blasticidin resistance cassette (BlastR). Lentiviral transduction yields GFP-positive cells, and subsequent infection with VSV-ΔL-dsRed particles yields GFP/dsRed-positive cells (yellow in merge). Addition of a 3CLpro inhibitor suppresses the accumulation of dsRed signal (inhibitor-off system). **d** Schematic of FluoroSpot Counter workflow to generate high-throughput data. **e** GC376 causes a dose-responsive suppression of dsRed signal (inhibitor-off) and a corresponding restoration of cell viability ($n = 3$ biologically independent replicates per condition with individual data points shown and average values represented by histogram bars).

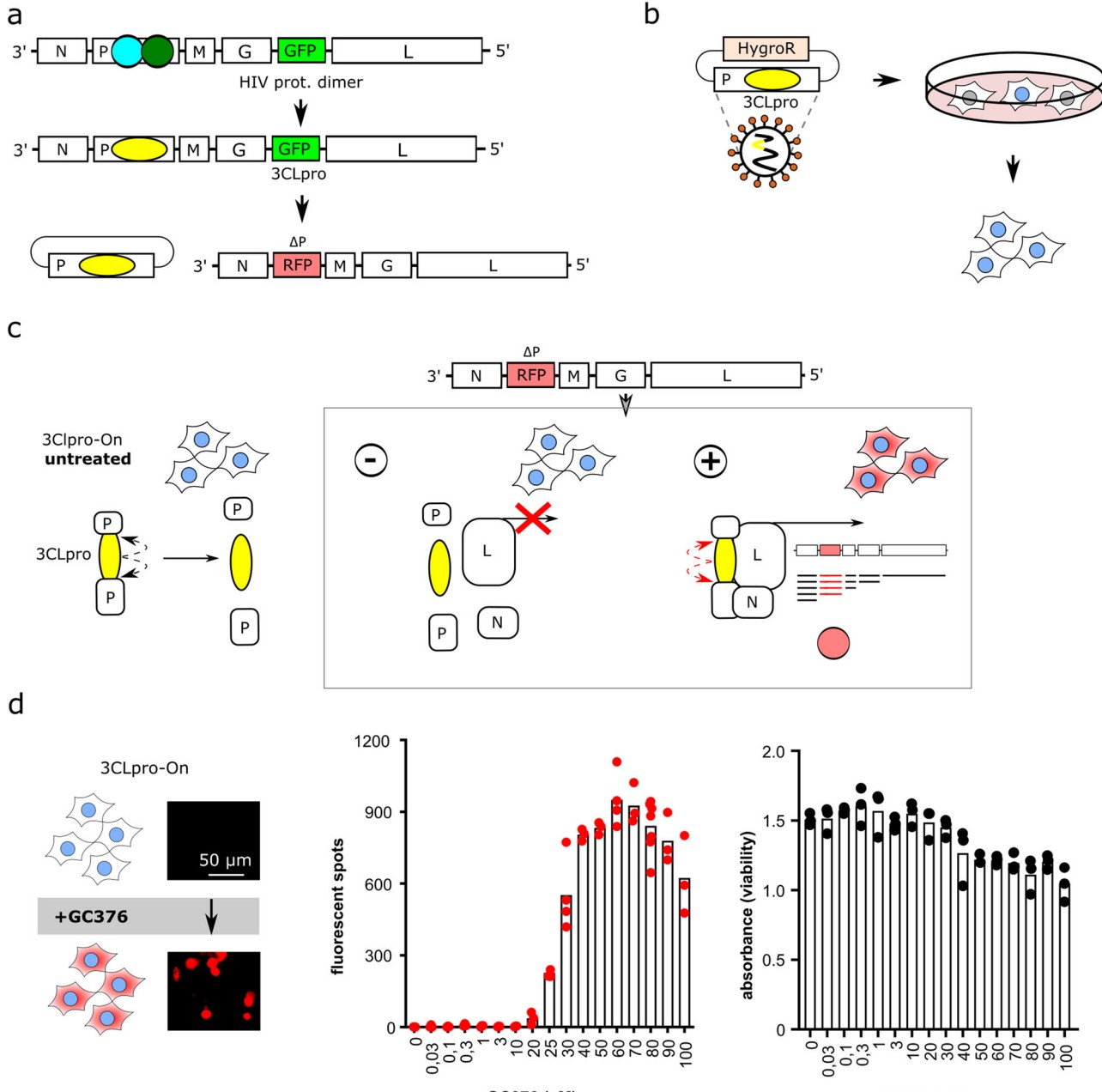

**Fig. 2 Design principles of the inhibitor-on system for quantifying 3CLpro activity and chemical inhibition. a–c** Schematics of the constructs leading to the current biosafety level 1 system consisting of VSV-ΔP-dsRed and Lenti-P:3CLpro with hygromycin-resistance cassette (HygroR). Co-expression of these two vectors in the same cells yields no virus replication due to VSV P protein auto-cleavage by 3CLpro. Treatment of these cells with a 3CLpro inhibitor restores P protein function and enables virus replication (inhibitor-on system). **d** GC376 causes a dose-responsive increase in dsRed signal. High compound concentrations cause a diminution of dsRed signal and a corresponding decrease in cell viability ($n = 3$ biologically independent replicates per condition with individual data points shown and average values represented by histogram bars).

these cleavage sites are similarly or differentially affected by SARS-CoV-2 3CLpro inhibition. We, therefore, created three additional constructs by changing the N-terminal, C-terminal, and N- and C-terminal cleavage site glutamine (Q) residues to uncleavable asparagines (N) (Fig. 3a).

As above, the parent constructs emitted little dsRed signal in the absence of a protease inhibitor GC376 (Fig. 3b). In contrast and as expected, the double Q-to-N mutant yielded full levels of dsRed signal regardless of the addition of GC376, because the VSV P protein could no longer be cleaved (Fig. 3b). Interestingly, the C-terminal Q-to-N cleavage site mutant showed inhibitor-on

kinetics similar to the parental construct, whereas the N-terminal Q-to-N cleavage site mutant demonstrated a strongly increased responsiveness to GC376 treatment (Fig. 3b). These results suggest that the N-terminal cis-cleavage event may be rapid and harder to inhibit and, if no longer necessary due to mutation, then the assay signal becomes fully dependent on inhibiting the slower C-terminal cis-cleavage reaction. This interpretation was supported by results with boceprevir, which prior studies have shown is a less potent 3CLpro inhibitor than GC376[32]. For instance, even high concentrations of boceprevir were unable to turn on the parental construct or the C-terminal cleavage site

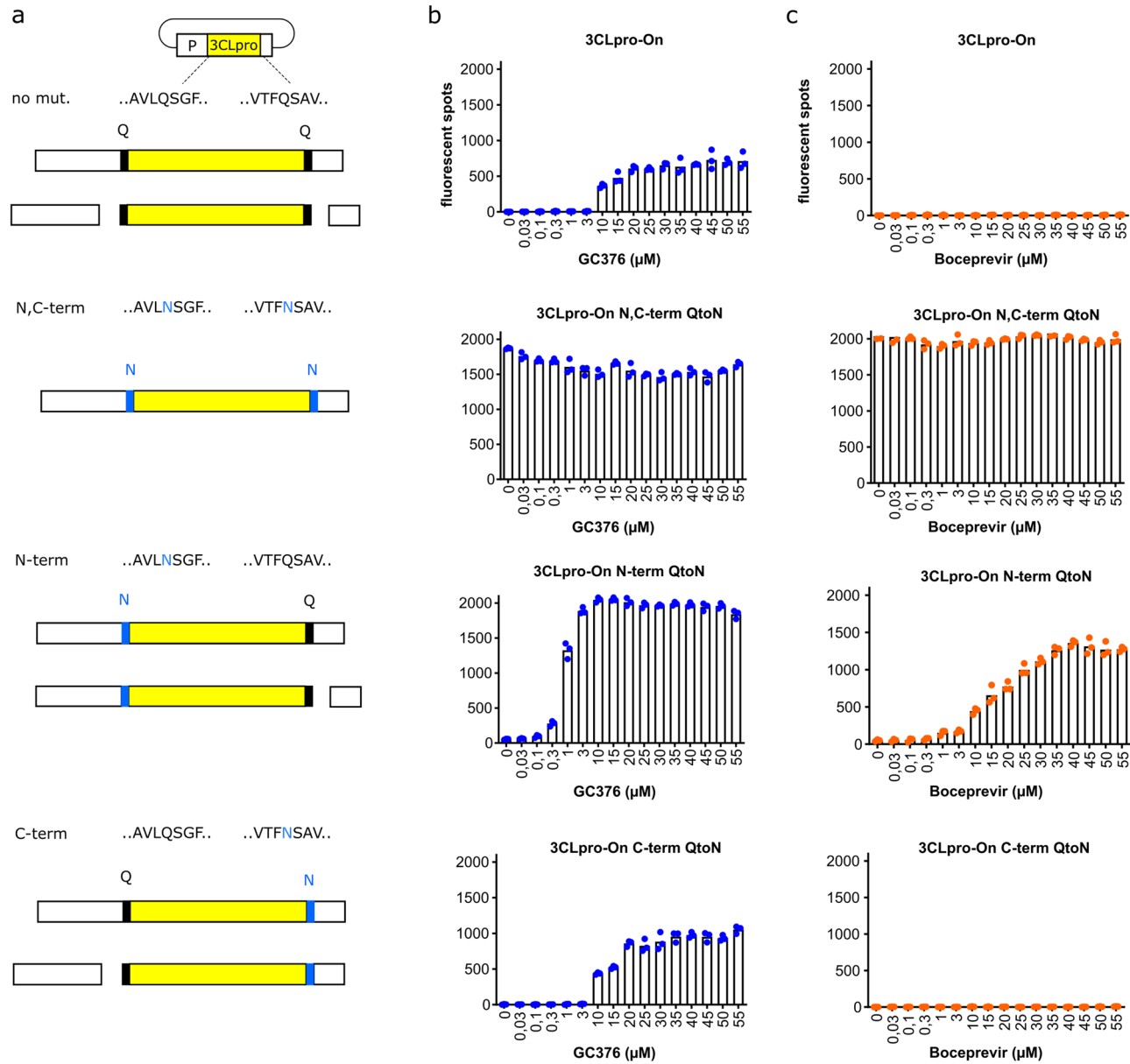

**Fig. 3 N- and C-terminal cis-cleavage mutants are inhibited differently by GC376 and boceprevir. a** Schematics of the four different constructs; "no mut." has functional N- and C-terminal cleavage sites leading to complete P protein disruption that can be recovered by protease inhibitor treatment; N,C-term has mutated N- and C-terminal cleavage sites and constitutive activity regardless of inhibitor treatment; N-term and C-term constructs have glutamine (Q) to asparagine (N) substitutions in the N- and C-terminal cleavage sites, respectively, which can be recovered differentially by protease inhibition. **b, c** GC376 and boceprevir dose-response experiments, respectively, with the constructs described in **a** ($n = 3$ biologically independent replicates per condition with individual data points shown and average values represented by histogram bars).

mutant, whereas this drug caused dose-responsive activation of the N-terminal cleavage mutant (Fig. 3c). Thus, the inhibitor-on system that relies solely upon C-terminal cleavage by 3CLpro appeared to provide the greatest sensitivity for inhibitor comparisons.

Complementary results were obtained with the inhibitor-off system, in which C-terminal cleavage inhibition by boceprevir compromised L protein function and caused a proportional decrease in dsRed signal. However, boceprevir did not inhibit N-terminal cleavage and therefore led to GFP separation from the construct as shown by an anti-GFP immunoblot (Supplementary Fig. 2).

**Assay signal stability of stably transfected cells over time**. To assess the reproducibility of the assay including fluorescent signal

stability over time, images were taken of inhibitor-containing wells over multiple passages of BHK21 cells expressing the parental inhibitor-on system and the cleavage site mutants described above. The fluorescent spot signals remained visible for more than ten passages spanning over 1 month (Supplementary Fig. 1g, h). A signal decrease in the C-terminal and N, C-terminal mutant constructs suggests that lower passage cells should be used for screening.

**Assay adaptability and application to different proteases**. The inhibitor-on system was developed originally for HIV-1 protease[19] and adapted here for SARS-CoV-2 3CLpro, thus suggesting that the assay may be further adapted for use with other viral proteases. To investigate this idea, we created a panel

 5

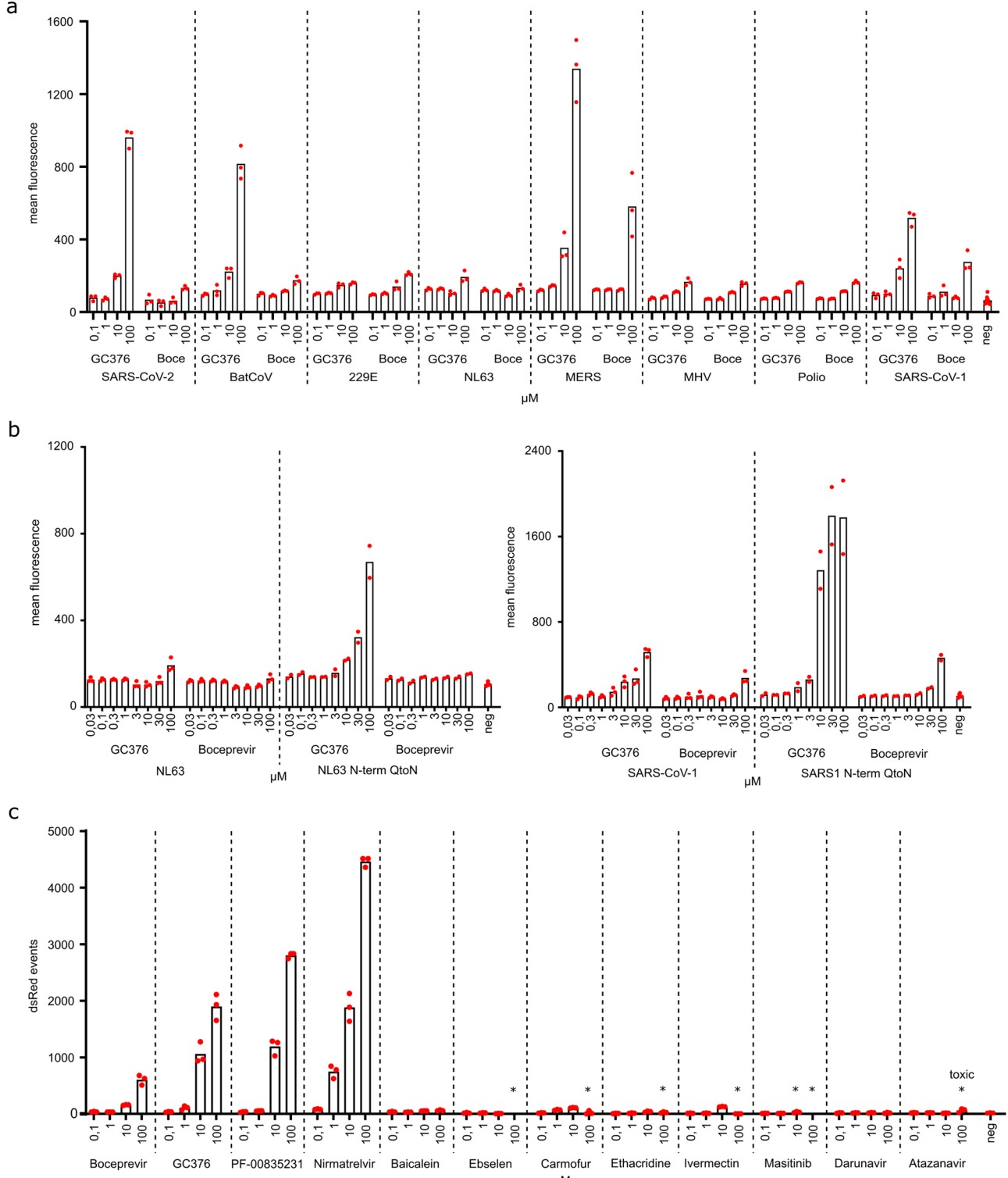

of constructs with a variety of different proteases and their cognate N- and C-terminal cleavage sites inserted into the VSV P protein (amino acid sequences in Supplementary Table 1). In the absence of protease inhibitor, these constructs all rendered the viral phosphoprotein inactive, as evidenced by low dsRed signal (Fig. 4a). Interestingly, like GC376 treatment of cells expressing SARS-CoV-2 3CLpro above, this compound caused a dose-responsive increase in dsRed signal of cells expressing the 3CLpro from Rousettus bat coronavirus HKU9 (batCoV), MERS, and SARS-CoV-1 (Fig. 4a). In comparison, proteases from human

coronaviruses 229E and NL63, mouse hepatitis virus (MHV), and poliovirus were less responsive to GC376 (maximal 2-fold increases from 100 nM to 100 µM). Boceprevir only had a significant effect on cells expressing the 3CLpro from MERS and SARS-CoV-1.

We next used the more sensitive C-terminal cleavage assay described above, to ask if greater signals might be achieved (Fig. 4b). Specifically, cells expressing either the NL63 or the SARS-CoV-1 3CLpro enzymes with the N-terminal cleavage site mutated from Q-to-N elicited significantly higher dsRed signal

**Fig. 4 Inhibition-on assay is adaptable to further proteases and helps dissect potent and proposed inhibitors. a** Seven proteases from close and distantly related viruses to SARS-CoV-2 (Rousettus bat coronavirus HKU9 (BatCoV), human coronaviruses 229E and NL63, MERS, mouse hepatitis virus (MHV), poliovirus, and SARS-CoV-1) were tested in 293T cells transfected with inhibition-on assay plasmids and treated with the compounds GC376 and boceprevir (abbreviated: Boce). Mean fluorescent intensity of positive cells was used to display read-outs (Supplementary Figs. 5 and 6). ($n = 3$ biologically independent replicates per condition with average values represented by histogram bars; negative control (neg) = average of transfected cells of each construct without inhibitor; dashed lines separate different constructs). **b** N-terminal cis-cleavage glutamine to asparagine mutants of the same seven proteases were tested for increased susceptibility to compounds. NL63 and SARS-CoV-1 showed increased response. (Wild-type: $n = 3$ biologically independent replicates per condition with average values represented by histogram bars; N-term-Q-to-N: $n = 2$ biologically independent replicates; negative control (neg) = transfected cells of each construct without inhibitor; dashed lines separate different constructs). **c** A panel of compounds was tested via FACS. Red fluorescent live singlet cell events were chosen to display read-outs (Supplementary Figs. 5 and 6). An asterisk above the compound concentrations (*) indicates visible toxicity (Supplementary Fig. 4). ($n = 3$ biologically independent replicates per condition with average values represented by histogram bars; negative control (neg) = infected cells without inhibitor; dashed lines separate different inhibitors).

indicative of active virus replication. Importantly, dsRed signal became apparent at lower inhibitor concentrations, and higher signals were achieved overall. Future studies with this inhibitor-on system may be able to map the important amino acids for antiviral compound binding (see also Supplementary Fig. 3 for amino acid alignments).

**Assay application to inhibitor testing**. A number of compounds have been reported thus far in the literature as SARS-CoV-2 3CLpro inhibitors. Some such as baicalein[33], ebselen[4], carmofur[34], ethacridine[35], ivermectin[36], and masitinib[37] are derived from compound repurposing attempts and more recent compounds such as PF-00835231[38] and PF-07321332/nirmatrelvir[39] have been optimized for SARS-CoV-2 3CLpro. To compare activities in the inhibitor-on N-terminal Q-to-N mutant system described here, all of these compounds were tested in parallel at 0.1, 1, 10, and 100 µM alongside the HIV-1 protease inhibitors darunavir and atazanavir as negative controls (Fig. 4c). Interestingly, only GC376, boceprevir, PF-00835231, and the recently approved compound PF-07321332/nirmatrelvir significantly inhibited 3CLpro and enabled virus replication and dsRed signal accumulation. Moreover, several compounds caused cytotoxicity at higher concentrations (asterisked in Fig. 4c; also see live-cell reductions in Supplementary Fig. 4). These results demonstrated that the assay described here can be useful for compound triage, identification, and/or optimization.

**Adaptation of inhibition-on assay to bioluminescence**. To facilitate future application of our assays in high-throughput screening efforts, we adapted the inhibition-on assay to bioluminescence by replacing dsRed as viral transgene with a firefly luciferase (Fig. 5a). We tested 293T cells, which were transiently transfected with the SARS-CoV-2 inhibition-on plasmid, with the Pfizer compounds PF-00835231 and PF-07321332/nirmatrelvir. Both compounds produced an effect proportional to their concentration (Fig. 5b). Furthermore, we performed dose responses with GC376, boceprevir, and PF-07321332/nirmatrelvir in BHK21 cells with a stable integration of bat coronavirus HKU9 or MERS main proteases inhibition-on constructs (Fig. 5c). The HKU9 3CLpro did not respond to boceprevir, but to GC376 and nirmatrelvir, whereas MERS 3CLpro was inhibited by boceprevir, GC376, and nirmatrelvir.

**Discussion**
In this manuscript, we describe VSV-based cellular assays to assess SARS-CoV-2 3CLpro activity and inhibitors thereof. These protease assays are either loss-of-signal or gain-of-signal, depending on their design. First, we tested SARS-CoV-2 3CLpro, then expanded to close and distantly related virus proteases. By mutating the two cis-cleavage sequences of the proteases in the

gain-of-signal or inhibition-on system, we observed that N-terminal cleavage was more difficult to inhibit and therefore C-terminal cleavage assays were more sensitive to inhibition. Lastly, the inhibitor-on system was used to compare a panel of reported inhibitors and showed that only a small subset (4/12) were active in the system described here including the recently FDA-approved compound PF-07321332 (nirmatrelvir).

Most previous SARS-CoV-2 3CLpro cellular and biochemical assays elicit signals that are reduced by chemical protease inhibitors. In such loss-of-signal assays, compounds of interest can potentially disturb components of the assay, thereby non-specifically decreasing the read-out[40]. For example, compound toxicity influences cell viability and, therefore, virus titer, obscuring a direct compound effect on virus replication. This issue was improved by introducing GFP into recombinant SARS-CoV-2[41], whereby compound and virus-induced cell death can be distinguished by the absence or presence of GFP, respectively. Similarly, the inhibition of a purified enzyme can lead to false hits, e.g., by oxidation or denaturation of the enzyme[40]. Several studies using purified 3CLpro[8,36,42,43] report inhibitory activity without using a counter assay to control for such nonspecific inhibition. Furthermore, if test compounds are autofluorescent, they can mimic the substrate fluorescence and obscure an inhibitory effect.

To address the nonspecific decrease of read-outs by compound toxicity, our assays comprise a fluorescent signal able to discriminate between actual decrease of signal and unspecific decrease due to cell death. This is an advantage of our 3CLpro inhibitor-off construct over live SARS-CoV-2-based assays. Compared to recombinant SARS-CoV-2-GFP, our assays are more practical and safer, because they require only biosafety level 1 facilities. In contrast to biochemical methods, our systems have non-regulatable counter assays, which is a desirable feature to distinguish unspecific inhibition by a compound of any other assay component[40]. Our inhibitor-on approach is a gain-of-signal assay, which overcomes some of the limitations inherent to negative read-out assays[40].

Previously described methods to detect 3CLpro activity based on cells are variants of the so-called "Flip-GFP"[44], quenched GFP[45], and Src-M$^{pro}$-Tag-eGFP[46] systems. In the first method, the typical barrel structure of GFP is split into two parts and the smaller part is contorted, which lowers its fluorescence yield. When 3CLpro cuts, the contortion is alleviated, increasing Flip-GFPs fluorescent signal. As with other assays, the activity measured with this method is trans-cleavage, meaning the protease dimer is already formed, fully active, and cleaves a non-adjacent sequence. However, as described by the designers of the Src-M$^{pro}$-Tag-eGFP assay, Flip-GFP assay has a very high background level in the absence of a protease activity[46]. The second method uses a GFP version with a protease-cleavable quenching sequence, which can also be cleaved in trans by the 3CLpro.

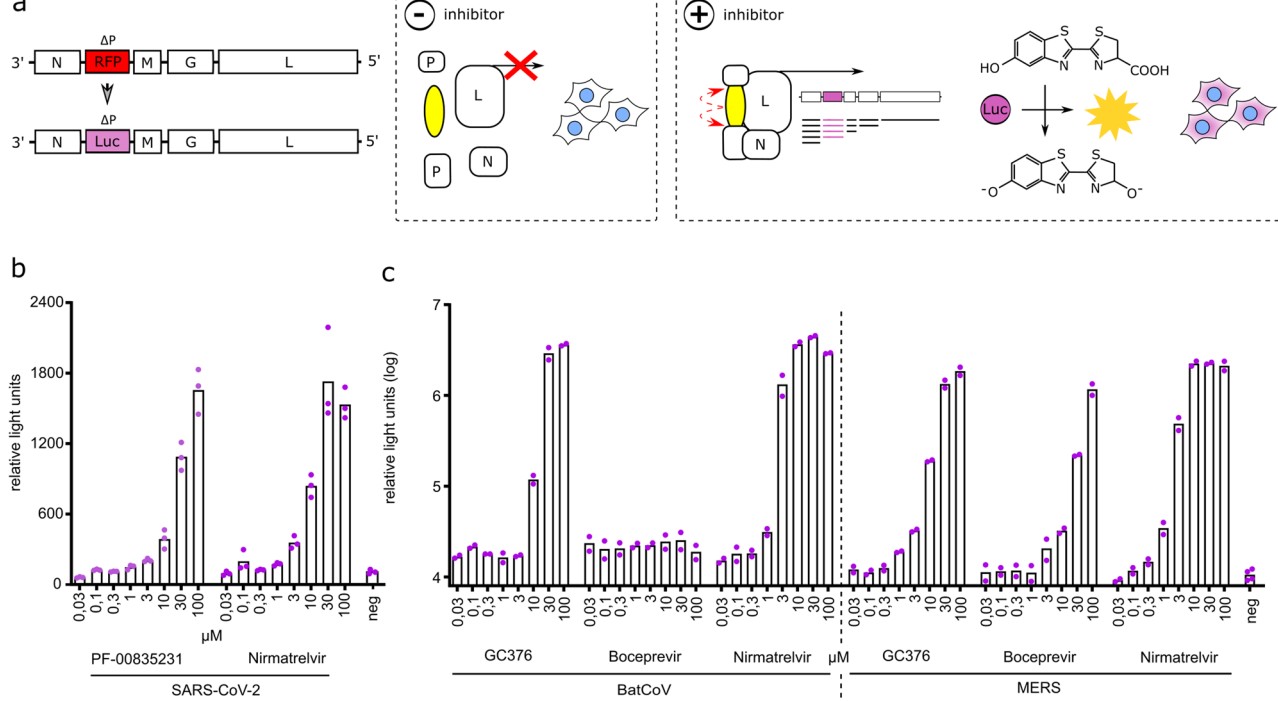

**Fig. 5 Adaptation of inhibition-on assay to bioluminescence. a** The red fluorescent protein (RFP) dsRed in VSV-ΔP was replaced with firefly luciferase, generating VSV-ΔP-Luc. Similar to the previous inhibition-on assay, addition of an inhibitor facilitates viral replication and gene expression. (dashed boxes separate schematics of molecular mechanism occurring in untreated (− protease inhibitor) versus treated (+ protease inhibitor) cells). **b** 293T cells were transiently transfected with the SARS-CoV-2 inhibition-on plasmid and treated with either PF-00835231 or PF-07321332/nirmatrelvir and VSV-ΔP-Luc. (n = 3 biologically independent replicates per condition with average values represented by histogram bars; negative control (neg) = average of transfected cells of each construct without inhibitor). **c** BHK21 stably transduced with bat coronavirus HKU9 (BatCoV) and MERS main proteases were tested with GC376, boceprevir, and PF-07321332/nirmatrelvir. (n = 2 biologically independent replicates per condition with average values represented by histogram bars; negative control (neg) = average of transfected cells of each construct without inhibitor; dashed lines separate differently transduced cells).

When this tag is cleaved off, GFP increases its fluorescence yield. The Src-M$^{pro}$-Tag-eGFP uses a mechanism, whereby a Src-M$^{pro}$-Tag-eGFP polyprotein leads to increasing GFP fluorescence by the addition of protease inhibitory compounds.

With the constructs in this study, cis-cleavage activity is a requirement. Consequently, an inhibitor that is only active after the formation of the mature dimer should not be detected by our assays. Previous assays assess trans-cleavage. Therefore, counter-screening inhibitors with both kinds of assay could be an interesting avenue to detect effective inhibitors.

In the course of the SARS-CoV-2 pandemic, many substances were proposed as potential inhibitors. To assess compound potency in a cellular environment, we tested a panel of reported compounds, namely baicalein, ebselen, carmofur, ethacridine, ivermectin, and masitinib as well as HIV protease inhibitors darunavir and atazanavir as controls. For most of those compounds, we did not find a strong activity but toxic effects at higher micromolar concentrations. Masitinib showed the strongest toxicity of all. We, therefore, propose that these compounds are either still precursors to inhibitors with only mild activity, don't penetrate the cell membrane, and/or act nonspecifically, for example, via influencing the cellular transcriptome or toxicity. However, four compounds indeed showed potent inhibitory activity, including nirmatrelvir, which has already achieved marketing authorization in the US.

A total of 9 viral proteases show activity with sensitivities in line with previous reports in the system described here. Interestingly, MERS 3CLpro sensitivity to GC376 was drastically lower than SARS-CoV-2 3CLpro in a biochemical assay (i.e., mature enzyme)[47], but was very sensitive in our assay. We, therefore,

speculate that monomeric/immature 3CLpro could have different sensitivities to inhibitors than mature enzymes.

Given the platforms flexibility, it is anticipated to be adjustable to newly emerging coronavirus vaccine- and immune-escape variants as well as other protease-dependent viruses. These assays, therefore, provide a rapidly adaptable protease inhibitor screening tool to discover compounds or repurpose existing compounds in future outbreaks.

## Methods

**Screening assay generation.** Screening assays are based on the replication machinery of vesicular stomatitis virus (VSV). Two-component systems were based on replication-incompetent VSVΔL-dsRed and VSVΔP-dsRed[27] as the viral components and transduced 293T (American Type Culture Collection, Manassas, VA) or BHK21 (American Type Culture Collection, Manassas, VA) cells. 293T cells were cultured in Dulbecco's Modified Eagle Medium (DMEM) (Lonza, Basel, Switzerland) supplemented with 10% FCS (Invitrogen, Carlsbad, California, USA), 1% P/S (PAA Laboratories, Pasching, Austria), 2% glutamine (PAA Laboratories), 1x sodium pyruvate (Gibco, Carlsbad, California, USA), 1x non-essential amino acids (Gibco). BHK21 cells were cultured in Glasgow minimum essential medium (GMEM) (Lonza) supplemented with 10% fetal calf serum (FCS), 5% tryptose phosphate broth, 100 units/ml penicillin and 0.1 mg/ml streptomycin (P/S) (Gibco).

First, regulatable constructs were cloned into VSV-GFP[48,49]. A VSV-GFP variant coding 3CLpro-Off was cloned by Gibson assembly (NEB, Ipswich, USA)[50]. The intergenic region between GFP and L was removed by restriction enzyme digestion of sites close to the end of GFP (MscI) and after the beginning of L (HpaI). Missing GFP and L parts were PCR amplified and overlapping sequences to the vector and SARS-CoV-2 protease inserted by primer pairs GFP-34bp-before-MscI-for / GFP-prot-rev and prot-L-for / L-33bp-after-HpaI-rev, respectively (Table 1). The SARS-CoV-2 protease sequence was retrieved from cDNA of purified virus isolates. The sequence of the SARS-CoV-2 protease corresponds to the Wuhan-1 isolate sequence (NCBI Reference Sequence: NC_045512.2, Supplementary Table 1). An additional restriction site (BbvCI) was introduced in

**Table 1 cloning primer for VSV vectors.**

| Name | Sequence (5'-3' direction) |
|---|---|
| prot-for | ATCACCTCAGCTGTTTTGCAG |
| prot-rev | GATTGTTCTTTTCACTGCACT |
| GFP-34bp-before-MscI-for | ATCTGCACCACTGGAAAGCTC |
| GFP-prot-rev | CTGCAAAACAGCTGAGGTGATCTTGTACAGCTCGTCCATGCC |
| prot-L-for | AGTGCAGTGAAAAGAACAATCATGGAAGTCCACGATTTTGAG |
| L-33bp-after-HpaI-rev | GATGTTGGGATGGGATTGGC |
| IGR-luciferase-for | CAGATATCACGCTCGAGGCAATTGCGCGCTAGCTATGAAAAAAACTAACAGATATCACCATGGAAGATGCCAAAAACATTAAG |
| Luciferase-cut1-rev | GCCATTTTTCTAAAACCACTCTGCAAAACAGCTGAGGTGATCACGGCGATCTTGCCGCCC |
| (GGSG)₃-prot-for | GGCGGAAGCGGCGGAGGGAGCGGGGGCGGGAGCGGAATCACCTCAGCTGTTTTGCAG |
| (GGSG)₃-prot-rev | GCCGGATCCACCGCCTGAGCCGCCTCCGGACCCTCCGATTGTTCTTTTCACTGCACT |
| N-35nt-before-BstZ17l-for | GCAAGGAATGCCCGACAGCC |
| P-GGSG-rev | GCTCCCTCCGCCGCTTCCGCCATCTGATACTGCTTCTGATTGG |
| GGSG-P-for | GGCTCAGGCGGTGGATCCGGCGTTTGGTCTCTCTCAAAGACAT |
| P-35nt-after-Xbal-rev | CGTCCGTCACCTCCGACAGAG |
| ΔP-Luc-for | CTAACAGATATCGAATTCCTGCAGCCCGGGGGGATCCACCGGTCGCCACCATGGAAGATGCCAAAAACATTAAG |
| ΔP-Luc-rev | GGGATAACACTTAGATCGTGATATCGTTACTTTTTTTCATAGTGCGGCCGCTACACGGCGATCTTGCCGC |

the N-terminal protease recognition sequence to facilitate further cloning. We used this BbvCI (NEB) site together with NheI (NEB) to remove GFP and introduce a firefly luciferase, thereby generating VSV-Luc-SARS-Prot-Off with primers IGR (intergenic region)-luciferase for and luciferase-cut1-rev (Table 1).

A VSV-GFP variant coding 3CLpro-On was also cloned via Gibson assembly. VSV-GFP was digested with BstZ17l and XbaI (NEB). Vector and SARS-CoV-2 protease overlapping fragments were PCR amplified with primers N-35nt-before-BstZ17l-for and P-GGSG-rev for upstream sequences of N and P that were omitted in the digestion and GGSG-P-for and P-35nt-after-XbaI-rev for downstream P sequence (Table 1). The SARS-CoV-2 protease was amplified from cDNA with primers (GGSG)₃-prot-for and (GGSG)₃-prot-rev (Table 1).

VSVΔP-Luciferase/Luc was cloned by digestion of the VSVΔP-dsRed plasmid with SmaI and NotI and Gibson assembly with a PCR on a firefly luciferase plasmid with primers ΔP-Luc-for and -rev.

Two-component system lentiviruses used in this study originate from blasticidin resistance encoding pLenti CMVie-IRES-BlastR (Addgene accession: #119863). An additional variant resistance gene was cloned into pLenti CMVie-IRES-BlastR, namely hygromycin (pLenti CMVie-IRES-HygroR). First, the blasticidin resistance was omitted by digestion of pLenti CMVie-IRES-BlastR with MscI and NotI (NEB). Then, hygromycin resistance was added by Gibson assembly of resistance genes plus vector overlapping sequences introduced by PCR.

Lentiviruses encoding regulatable 3CLpro-Off and -On switches were generated with Gibson assembly as follows. For 3CLpro-Off, first an L protein blasticidin lentivirus was cloned with primers LV-L-for and L-LV-rev (Table 2) to facilitate the cloning of the large fusion construct. This plasmid was then digested with NheI and HpaI. GFP, the SARS-CoV-2 protease and the N-terminal part of L omitted by HpaI digestion were replaced by a PCR on the full VSV-3CLpro-Off-GFP plasmid with primers LV-GFP-for and L-rev (Table 2). 3CLpro-On was amplified fully from VSV-3CLpro-On-GFP plasmid with primers LV-P-for and P-LV-rev (Table 2). Both constructs were deposited in GenBank with accession numbers ON262564 (3CLpro-Off blasticidin) and ON262565 (3CLpro-On hygromycin).

Hygromycin-resistance-based lentivirus vectors encoding variable proteases were cloned via Gibson assembly. First, proteases and adjacent cleavage sites were codon-optimized and synthesized by Integrated DNA Technologies (USA). For integration into inhibition-on constructs, they were amplified and extended with a flexible (GGSG)₃-linker sequence with primer pairs GGSG-*virus*-On-for and *virus*-GGSG-On-rev (Table 2). The (GGSG)₃-linker sequence was used for a fusion PCR with N- and C-terminal phosphoprotein fragments generated by PCRs on previous phosphoprotein constructs with primer pairs LV-P-for with P-GGSG-rev and GGSG-P-for with P-LV-rev. The phosphoprotein fragments containing GGSG sequences and (GGSG)₃-linker sequence extended proteases were joined with a fusion PCR with primers LV-P-for and P-LV-rev. These fusion PCRs were then ligated into a hygromycin-resistance vector digested with NheI and PacI.

N-terminal glutamine to asparagine mutants of variable proteases were generated via Gibson assembly. Forward primer containing a (GGSG)₃-linker sequence and a Q-to-N mutation were designed for all proteases (Table 2). PCRs on existing constructs of variable proteases were performed with primers *virus*-On-N-term-QtoN-rev was paired and P-LV-rev. These fragments spanning the respective protease and the C-terminal phosphoprotein fragment were fused to the N-terminal phosphoprotein sequence via a fusion PCR and then ligated into a hygromycin-resistance vector digested with NheI and PacI.

**Lentiviral transduction**. Lentiviruses were generated by CaPO₄ transfection of lentiviral plasmids together with Gag-Pro-Pol and VSV glycoprotein[51]. Lentivirus containing supernatants were harvested 24 and 48 h after changing transfection medium containing chloroquine (~ 12 h after CaPO₄ transfection) and pooled. Pooled supernatants were used to perform spin-infection of $4 \times 10^5$ 293T or BHK21 cells per well in a 6-well plate at 1000 g and 37 °C with medium containing 8 µg/ml polybrene (Sigma, St. Louis, US)[52]. Two days after transduction, cells were split selected either by 12 µg/ml for 293T and 12 µg/ml for BHK21 blasticidin (InvivoGen, France) and 400 µg/ml for 293T and 600 µg/ml for BHK21 hygromycin (InvivoGen, France).

**Screening assay with FluoroSpot read-out**. Compounds were screened in a 96-well format. Ten thousand 293T or BHK21 cells expressing either a regulatable construct or native VSV proteins were seeded per well (cell number can be adjusted up to 20.000 cells per well for toxic compounds). Four hours after seeding, compounds and virus (multiplicity of infection, MOI: 0.01 of VSV-ΔP-RFP, MOI: 0.1 of VSV-ΔL-RFP) were added to wells. After 40 h, supernatants were removed, and fluorescent spots counted in a Fluoro/ImmunoSpot counter (CTL Europe GmbH, Bonn, Germany) with the manufacturer-provided software CTL switchboard 2.7.2. 90% of each well area was scanned concentrically to exclude reflection from the well edges, and counts were normalized to the full area. Automatic fiber exclusion was applied while scanning. The excitation wavelength for RFP was 570 nm, the D_F_R triple band filter was used to collect fluorescence. Manual quality control for residual fibers was also performed. In parallel, plates with the same compound treatment scheme were incubated with 20 µL of 5 mg/ml Thiazolyl Blue Tetrazolium Bromide / MTT (SIGMA) for 4 h, then lysed with 0.1 g sodium dodecyl sulfate/ml 0.01 M HCl over night with gentle shaking. MTT absorbance was measured at 550 nm (main absorbance) and 655 (base absorbance to substract).

Alternatively, spot counts were performed with a BZ-X810 All-in-One fluorescence microscope from Keyence (Ōsaka, Japan) or a Cytation|1 Imaging reader from BioTek (Vermont, USA). Exemplary read-outs are displayed in Supplementary Fig. 7.

**Screening assay with fluorescence-activated cell scanning (FACS) read-out**. Transiently transfected 293T cells: $4 \times 10^5$ cells per 6-well were seeded 1 day prior to transfection. Plasmids were transfected with TransIT®-LT1 transfection kit (Mirus Bio LLC, Madison, WI, USA) with 800 ng plasmid DNA and the recommended 1:3 µg-DNA: µl-reagent ratio. Six to twelve hours after transfection, 293T cells were split and 15.000 cells seeded in 96-well plates in 50 µl medium. Compound and virus (MOI 0.1) were added in 50 µl to reach desired concentrations. After 2 days, cells

**Table 2 cloning primer for lentivirus plasmids.**

| Name | Sequence (5'-3' direction) |
|---|---|
| LV-L-for | CTGTTTTGACCTCCATAGAAGATTCTAGAGCTAGCATGGAAGTCCACGATTTTGAG |
| L-LV-rev | GAGGGAGAGGGGCGGATCCCCTTAATTAATTAATCTCTCCAAGAGTTTTCCTC |
| LV-GFP-for | GACCTCCATAGAAGATTCTAGAGCTAGCATGAGCAAGGGCGAGGAACTG |
| L-33bp-after-HpaI-rev | GATGTTGGGATGGGATTGGC |
| LV-P-for | CTGTTTTGACCTCCATAGAAGATTCTAGAGCTAGCATGGATAATCTCACAAAAGTTC |
| P-LV-rev | GAGGGAGAGGGGCGGATCCCCTTAATTAACTACAGAGAATATTTGACTCTCGC |
| GGSG-BatCoV-On-for | GGCGGAAGCGGCGGAGGGAGCGGGGGCGGGAGCGGAAGCGTCGCCAGTGCTGCGC |
| BatCoV-GGSG-On-rev | GCCGGATCCACCGCCTGAGCCGCCTCCGGACCCTCCTCGAAACATAGATTGAAATTTACCTTG |
| GGSG-HCoV-229E-On-for | GGCGGAAGCGGCGGAGGGAGCGGGGGCGGGAGCGGATATCTTATGGCTCAACGCTCCAA |
| HCoV-229E-GGSG-On-rev | GCCGGATCCACCGCCTGAGCCGCCTCCGGACCCTCCAAACATGGATGTAGTCTTACCAGATTG |
| GGSG-HCoV-NL63-On-for | GGCGGAAGCGGCGGAGGGAGCGGGGGCGGGAGCGGAATCAGTTACAATAGTACCTTGCAAAGC |
| HCoV-NL63-GGSG-On-rev | GCCGGATCCACCGCCTGAGCCGCCTCCGGACCCTCCAAGCCCGAATATAACCTTTCCTG |
| GGSG-MERS-On-for | GGCGGAAGCGGCGGAGGGAGCGGGGGCGGGAGCGGATCAATCACTAGCGGTGTATTGC |
| MERS-GGSG-On-rev | GCCGGATCCACCGCCTGAGCCGCCTCCGGACCCTCCGTATGTTACTTTTCTTACACCGGAC |
| GGSG-MHV-A59-On-for | GGCGGAAGCGGCGGAGGGAGCGGGGGCGGGAGCGGATCAGTCACCACTTCATTTCTCC |
| MHV-A59-GGSG-On-rev | GCCGGATCCACCGCCTGAGCCGCCTCCGGACCCTCCTTTTATTACTCTAGTCCTTTTACTCTGCAG |
| GGSG-Polio-On-for | GGCGGAAGCGGCGGAGGGAGCGGGGGCGGGAGCGGAACCATTCGGACAGCAAAGGTAC |
| Polio-GGSG-On-rev | GCCGGATCCACCGCCTGAGCCGCCTCCGGACCCTCCAGGTCTCATCCACTGGATTTC |
| GGSG-SARS1-On-for | GGCGGAAGCGGCGGAGGGAGCGGGGGCGGGAGCGGAAGTATCACGTCTGCTGTGC |
| SARS1-GGSG-On-rev | GCCGGATCCACCGCCTGAGCCGCCTCCGGACCCTCCTTTGACTATTTTTTTGAACTTACCTTG |
| BatCoV-On-N-term-QtoN-for | GGCGGAAGCGGCGGAGGGAGCGGGGGCGGGAGCGGAAGCGTCGCCAGTGCTGCGCTCAACGCGGGTCTTACTC |
| HCoV-229E-On-N-term-QtoN-for | GGCGGAAGCGGCGGAGGGAGCGGGGGCGGGAGCGGAGTATCTTATGGCTCAACGCTCAACGCCGGCTTGCGC |
| HCoV-NL63-On-N-term-QtoN-for | GGCGGAAGCGGCGGAGGGAGCGGGGGCGGGAGCGGAATCAGTTACAATAGTACCTTGAACAGCGGACTG |
| MERS-On-N-term-QtoN-for | GGCGGAAGCGGCGGAGGGAGCGGGGGCGGGAGCGGATCAATCACTAGCGGTGTATTGAACAGTGGTTTGGTC |
| MHV-On-N-term-QtoN-for | GGCGGAAGCGGCGGAGGGAGCGGGGGCGGGAGCGGATCAGTCACCACTTCATTTCTCAACTCCGGGATAG |
| Polio-On-N-term-QtoN-for | GGCGGAAGCGGCGGAGGGAGCGGGGGCGGGAGCGGAACCATTCGGACAGCAAAGGTAAACGGACCAGGGTTC |
| SARS1-On-N-term-QtoN-for | GGCGGAAGCGGCGGAGGGAGCGGGGGCGGGAGCGGAAGTATCACGTCTGCTGTGCTCAACTCAGGCTTCAG |

**Table 3 Z-factor values, divided in differentiated (multiple Z-factors) and non-differentiated (single Z-factor) according to the day of read-out (D12 → D39).**

| Construct | Single Z-factor | Multiple Z-factors | | | | | |
|---|---|---|---|---|---|---|---|
| | D12→D39 | D12 | D15 | D18 | D20 | D23 | D39 |
| 3CLpro-On | 0,53 | 0,86 | 0,86 | 0,51 | 0,68 | 0,79 | 0,76 |
| 3CLpro-On N-term | 0,64 | 0,86 | 0,68 | 0,52 | 0,92 | 0,84 | 0,67 |
| 3CLpro-On C-term | 0,11 | 0,29 | 0,75 | 0,72 | 0,56 | 0,34 | 0,66 |
| 3CLpro-On N,C-term | Read-out is always positive, with or without the inhibitor (see Supplementary Fig. 1h) | | | | | | |

were detached with 0.05 % Trypsin-EDTA (Gibco) and transferred to a 96-well round-bottom plate (TPP Techno Plastic Products AG, Switzerland) for automatic sampling by fluorescence-activated cell scanning (BD LSRFortessa X-20). Stably transduced BHK21 cells: fifteen-thousand cells were seeded and immediately treated with compound doses and VSV-ΔP-RFP at MOI 0.01. After 2 days, cells were treated as described above and measured via FACS. Gates for live cells and singlets were applied before division in fluorescent and non-fluorescent 293T and BHK21 cells (Supplementary Figs. 5 and 6). Both mean fluorescence intensity (MFI) of singlets and dsRed fluorescent events of P1 are used to quantify read-out. MFI read-outs were more sensitive and were therefore used for transiently transfected cells, which generally showed a weaker response than stably transduced cells. Stably transduced cells treated with toxic compounds showed autofluorescence. Hence, MFI was less useful to distinguish actual signal from artefacts. Thus, we chose dsRed events to display compound cross-comparisons.

**Screening assay with luciferase read-out.** Transiently transfected 293T cells and stably transduced BHK21 cells were prepared as described above in FACS screening. After 2 days, BHK21 cells were lysed with Bright-Glo™ (Promega, Madison, USA) and measured with a SPARK bioluminescence reader from Tecan (Grödig, Austria). 293T cells were prepared with VivoGlo™ luciferin (Promega) and measured with a GloMax Explorer (Promega).

**Statistics and reproducibility**

*Reproducibility.* Sample sizes were chosen empirically based on experience from our previous studies. At least two biologically independent replicates were performed

per condition. Biologically independent replicates meaning distinct wells with the same conditions, not only multiple measurements of the same well. Although inter-assay variability of overall signal was observed (see also Supplementary Fig. 1g, h), intra-assay comparison distinguished compound potency reliably.

*Z-factor.* To validate basic metrics of the assay, we calculated Z-factors for each 3CLpro gain-of-function (On – N – C – N,C) expressing cell line. We observed that the most of the calculated Z-factor values are within the interval indicated by Zhang et al.[53]. for an excellent assay. In brief, threshold values for both positive and negative control values were calculated as described in the following. The negative threshold value is equal to the mean signal of the negative controls plus three times their standard deviation (SD), while the positive threshold value is equal to the mean signal of the positive controls minus three times their SD. Then, the difference between the thresholds has been calculated and defined as S, 'separation band'. We subsequently computed the absolute difference between the two means, defined as 'dynamic range', R. Ultimately, we calculated the Z-factor as the product of S/R. These results suggest the reliability of this cell-based assay (Table 3).

*Half maximal effective concentration ($EC_{50}$).* Given a cellular system as basis of $EC_{50}$ calculation, we expect the dynamic range of the assay to be greater than in a biochemical assay, where the amount of enzyme is stable. In an excess of potent compounds, protease-viral fusion proteins are constantly renewed and cells continue growth. Therefore, virus replicates continuously and produces more read-out through excess compound. At lower concentrations, compound molecules are depleted and signal plateaus.

*Effective concentrations were calculated for 3CLpro-On N-term FACS data.* GC376: 15.04 ± 7.58 SD (95% confidence interval 11.54–19.69); boceprevir: 20.83 ± 5.98 SD (95% confidence interval 17.74–24.44); PF-00835231: 11.10 ± 2.95 SD (95% confidence interval 10.53–11.71); nirmatrelvir: 10.83 ± 6.65 SD (95% confidence interval 8.532–13.72) (Fig. 4c). EC$_{50}$ calculations were performed with Graphpad Prism 8 according to the recommendations of the software's user guide. For each compound, the highest signal was defined as 100%, and values of all concentrations and replicates were normalized to this value. For every compound, we performed a nonlinear regression analysis, in particular the agonist-versus-response function. For EC$_{50}$ the constrain (defined in Graphpad Prism 8 as the constant F) was set to 50. Top and bottom values were defined as 0 and 100, respectively. The standard deviation was calculated with following formula:

$$SD = \sqrt{\frac{\sum(residual^2)}{n - K}} \qquad (1)$$

Residuals are defined as the vertical distance of the point from the fit or curve. K is the number of parameters fit by regression.

**Immunoblotting.** Samples for immunoblotting were collected from 293T cells expressing VSV-L. We used this replication-supporting cell line for the expression of sufficient fusion protein (GFP-3CLpro-L) in the non-active condition (+ PI) of VSV-3CLpro-Off. SDS-PAGEs of protein lysates were performed under reducing conditions, on an 8% polyacrylamide gel for VSV-GFP and VSV-3CLpro-Off constructs. Gels were run for 90 min at 100 Volt. Proteins were transferred to 0.45-µm nitrocellulose membranes (Whatman, Dassel, Germany) by using a tank blotting system (Bio-Rad, Hercules, CA, US). The blotting time was 80 min. Blotting buffer contained 15 % methanol. The membranes were blocked overnight with 1x PBS containing 5% skim milk and 0.1% Tween 20 (PBSTM). GFP was stained by a mouse antibody (clones 7.1 and 13.1; Roche, Basel, Switzerland) diluted 1:1000 in PBSTM and a rabbit GFP/YFP antibody produced by Stephan Geley (conc.: 0.3 mg/mL) diluted in 1:2000 in PBSTM. β-Actin was stained with a monoclonal mouse anti-Actin antibody, A5441-.5 ML from SIGMA diluted 1:1000 in PBSTM. Blots were not stripped prior to β-Actin staining. PageRuler$^{TM}$ Pre-stained Protein ladder 26616 (ThermoFisher, Massachusetts, USA) was used as marker.

**Fluorescence microscopy.** 10$^5$ BHK21 VSV-L expressing cells were seeded in glass-bottom dishes with four chambers (ibidi GmbH, Gräfelfing, Germany) 8 h before infection. Cells were infected with an MOI of 1. Single images were acquired up to 16 h after infection at 37 °C using a 63X/NA1.4 objective on an automated live cell imaging Zeiss Axiovert 200 M microscopy equipped with a Sola light engine LED light source (Lumencor, Visitron Systems GmbH, Puchheim, Germany), a pco.edge 4.2 scMOS camera (PCO AG, Kelheim, Germany), controlled by VisiView software (Visitron). Exposure times were 200 ms for GFP and 10 ms for phase contrast.

**Reporting summary.** Further information on research design is available in the Nature Research Reporting Summary linked to this article.

## Data availability

All pertinent data to support this study are included in the manuscript and supplementary material. Unedited western blots are depicted in Supplementary Fig. 8 and data used to compile graphs are provided in Supplementary Data 1. If required, further data supporting the findings are available from the corresponding authors upon reasonable request.

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

## Acknowledgements

We thank Franziska Podesser, MSc. and Dr. Hadwiga Heilmann for useful discussions and Ivan Ploner, MSc. for technical assistance. E.H. is a recipient of a DOC Fellowship of the Austrian Academy of Science.

## Author contributions

E.H. conceived the initial concept. E.H., S.G., A.V., and F.C. designed the experiments. E.H. conceived the cloning strategies. E.H. generated recombinant viruses. A.V. previously established a reliable lentiviral transduction system in our institute. S.A.M. provided protease constructs and experimental expertize. E.H., S.G., and F.C. performed experiments. E.H., S.G., B.R., D.v.L., and R.S.H. wrote the manuscript. D.v.L. and R.S.H. provided the infrastructure and funding to the project. D.v.L., B.R., and R.S.H. provided supervision of E.H. The co-authors F.C., S.G., S.A.M., and A.V. equally contributed to this work. Their names in the title pages author line up are listed alphabetically. All authors read and approved the final manuscript.

## Competing interests

D.v.L. is founder of ViraTherapeutics GmbH. D.v.L serves as a scientific advisor to Boehringer Ingelheim and Pharma KG. E.H. and D.v.L have received an Austrian Science Fund (FWF) grant in the special call "SARS urgent funding". RSH is the Margaret Harvey Schering Land Grant Chair for Cancer Research, a Distinguished McKnight University Professor, and an Investigator of the Howard Hughes Medical Institute. All other authors declare no competing interests.
