## [Peer Review File · Communications Biology]

Reviewers' comments:

Reviewer #1 (Remarks to the Author):

The manuscript provided by E Heilmann et al. describes the development of a novel, cell-based, BSL-1 suitable antiviral assay for the screening of SARS-CoV-2 protease inhibitors. In addition, the authors show how a modified version of the assay can be used to answer mechanistic questions, particularly to distinguish between protease inhibitors with N- and C-terminal cleavage activity. The manuscript, a logical extension on the author's previous work on HIV protease on/off switches to regulate VSV, is concise, well-conceived and well-written. The methods section provides sufficient detail to allow other researchers in the field to reproduce the findings.

Since the main claim of the manuscript is the development of a novel coronavirus protease inhibitor screening assay, additional information on the robustness (signal/background ratio of photometric read-out, Z' factor of assay) and reproducibility of the assay (signal of stably transfected cells over 10-20 passages; repeat ($n \geq 3$) of GC376 dose response curve with EC50 and EC90 calculation, mean \pm standard deviation) would strengthen the paper. A second limitation of the paper is the lack of further validation with other well described SARS-CoV-2 protease inhibitors, particularly PF-07304814/PF-00835231 (Boras B et al., <https://doi.org/10.1101/2020.09.12.293498>). PF-00835231 is commercially available from multiple vendors and is considered a much more specific inhibitor of the SARS-CoV-2 3CLpro, while GC376 displays significant off-target activity against host proteases such as cathepsin L. Some additional corrections to consider: line 30: Boceprevir is an HCV inhibitor (not HIV); line 196: please reconsider the statement of GC376 as best currently known and available SARS-CoV-2 protease inhibitor in light of the Boras et al publication mentioned above or, for example, Vandyck K et al., doi: 10.1016/j.bbrc.2021.03.096).

Given the ongoing Covid-19 pandemic and the urgent need for novel antiviral drugs to combat SARS-CoV-2, the BSL-1 suitable, cell-based screening assay described in this manuscript should accelerate the discovery of novel SARS-CoV-2 protease inhibitors. I recommend publication of the manuscript if the concerns mentioned above can be addressed by the authors.

Reviewer #2 (Remarks to the Author):

The manuscript by Heilmann et al. outlines a series of cell-based assays to measure the ability of small molecule protease inhibitors to block viral proteases auto-processing events. In the example case present in the manuscript, the system is used to monitor inhibition of specific cis-cleavage events mediated by the viral protease 3CLpro SARS-CoV-2. The main premise is that this assay allows assessment of inhibitor potency specifically towards early cis cleavage events. It also provides information about the overall cell permeability of the compounds as they need to get inside cells to block this protease activity. Finally, the authors have developed two variations of the assay, one which they call 'on' and the other 'off'. These assays allow monitoring of both C and N terminal cleavage events of the polyprotein.

In general, this is a nice idea and seems to be a useful assay for SARS-CoV-2 Mpro. It may be more useful than simply screening compounds against the purified protease using small peptide substrates as a readout. Furthermore, there is the potential for this strategy to be suitable for use with other viral proteases such as SARS-CoV-2 PLpro and proteases from other viruses. However, there is only data in this paper for Mpro. While these are nice assays, they still potentially lack relevance to the actual polyprotein cleavage events that take place inside an infected cell. The constructs only represent a fragment of the actual poly proteins and the use of a GFP tag creates an artificial substrate for the protease. It is therefore not clear if the cleavages that are being measured in this assay are actual indicative of the real cis cleavages that take place in the virally infected cell. In addition, the expression of the polyprotein may result in different levels of protein compared to what is present inside an infected cell. There is also likely difference in the localization and cellular compartmentalization of the polyprotein that takes place in an actual infected cell compared to the fragment of the protein expressed inside a reporter cell line. Therefore, it is possible that compounds that show activity in this assay may not be effective at blocking cis-cleavages in a cell infected with SARS-CoV-2. Finally, the study seems premature and

lacking in validation beyond a known inhibitor and a single protease target. There are no new screens performed and no new biology that results from this study. It therefore seems to lack the novelty, impact and new insight required for publication in Nature Communications.

In addition, there are some minor issues that should be addressed in future versions of the manuscript.

In figure 1 it is not clear what the right and left panels of cell images represent in each set of images. Are they the different time points before and after addition or removal of inhibitors? It does not seem that it is but it is not clear what is changing in the panels.

In figure 3 there should be labels for the X-axis for all graphs, not just the top graphs. The label for the bar graph X- axis is also missing in panel e.

Reviewer #1

The manuscript provided by E Heilmann et al. describes the development of a novel, cell-based, BSL-1 suitable antiviral assay for the screening of SARS-CoV-2 protease inhibitors. In addition, the authors show how a modified version of the assay can be used to answer mechanistic questions, particularly to distinguish between protease inhibitors with N- and C-terminal cleavage activity. The manuscript, a logical extension on the author's previous work on HIV protease on/off switches to regulate VSV, is concise, well-conceived and well-written. The methods section provides sufficient detail to allow other researchers in the field to reproduce the findings.

Since the main claim of the manuscript is the development of a novel coronavirus protease inhibitor screening assay, additional information on the robustness (signal/background ratio of photometric read-out, Z' factor of assay) and reproducibility of the assay (signal of stably transfected cells over 10-20 passages; repeat ($n \geq 3$) of GC376 dose response curve with EC50 and EC90 calculation, mean \pm standard deviation) would strengthen the paper.

A second limitation of the paper is the lack of further validation with other well described SARS-CoV-2 protease inhibitors, particularly PF-07304814/PF-00835231 (Boras B et al., <https://doi.org/10.1101/2020.09.12.293498>). PF-00835231 is commercially available from multiple vendors and is considered a much more specific inhibitor of the SARS-CoV-2 3CLpro, while GC376 displays significant off-target activity against host proteases such as cathepsin L.

We thank referee #1 for his/her constructive comments. As suggested, we have calculated the Z-factor and EC50 values. These metrics are summarized in **table 3**. However, as we also discuss in the methods section, EC50/90 values are generally higher in cell-based assays than in standard biochemical assays. The constant renewal of protease fusion proteins also drives signal increases. Therefore, signals are expected to plateau later than in a biochemical assay with a fixed amount of enzyme. Cellular assays therefore have a greater dynamic range and calculating EC50/90 values is only useful for inhibitor cross-comparisons for our system.

We purchased PF-00835231 and PF-07321332/nirmatrelvir and tested them successfully in different settings (see **Figures 4c, 5b and 5c**). We also tested compounds that were proposed to be inhibitors in some studies but were not reproduced in others. The compounds baicalein, ebselen, carmofur, ethacridine, ivermectin, masitinib, darunavir, and atazanavir did not have activity in our assay, but were toxic (**Figure 4c and Figure S4**).

Some additional corrections to consider: line 30: Boceprevir is an HCV inhibitor (not HIV); line 196: please reconsider the statement of GC376 as best currently known and available SARS-CoV-2 protease inhibitor in light of the Boras et al publication mentioned above or, for example, Vandyck K et al., doi: 10.1016/j.bbrc.2021.03.096).

We have made these corrections. Thank you.

Given the ongoing Covid-19 pandemic and the urgent need for novel antiviral drugs to combat SARS-CoV-2, the BSL-1 suitable, cell-based screening assay described in this manuscript should accelerate the discovery of novel SARS-CoV-2 protease inhibitors. I recommend publication of the manuscript if the concerns mentioned above can be addressed by the authors.

Thank you.

Reviewer #2

The manuscript by Heilmann et al. outlines a series of cell-based assays to measure the ability of small molecule protease inhibitors to block viral proteases auto-processing events. In the example case present in the manuscript, the system is used to monitor inhibition of specific cis-cleavage events mediated by the viral protease 3CLpro SARS-CoV-2. The main premise is that this assay allows assessment of inhibitor potency specifically towards early cis cleavage events. It also provides information about the overall cell permeability of the compounds as they need to get inside cells to block this protease activity. Finally, the authors have developed two variations of the assay, one which they call 'on' and the other 'off'. These assays allow monitoring of both C and N terminal cleavage events of the polyprotein.

In general, this is a nice idea and seems to be a useful assay for SARS-CoV-2 Mpro. It may be more useful than simply screening compounds against the purified protease using small peptide substrates as a readout. Furthermore, there is the potential for this strategy to be suitable for use with other viral proteases such as SARS-CoV-2 PLpro and proteases from other viruses. However, there is only data in this paper for Mpro.

Thank you for this thoughtful assessment of our work. As recommended, we now demonstrate the modularity of our assay by testing 7 additional proteases, with SARS1, MERS, and the HKU9 bat coronavirus protease also showing susceptibility to GC376 (**Figure 4a**). We also show for a subset of these proteases that N-terminal cleavage site mutant constructs demonstrate a greater sensitivity to inhibition (**Figure 4b**).

While these are nice assays, they still potentially lack relevance to the actual polyprotein cleavage events that take place inside an infected cell. The constructs only represent a fragment of the actual poly proteins and the use of a GFP tag creates an artificial substrate for the protease. It is therefore not clear if the cleavages that are being measured in this assay are actual indicative of the real cis cleavages that take place in the virally infected cell.

First, please note that in all instances the substrate for the protease is its cognate/natural cleavage site built into a virus polyprotein encoding in addition a reporter construct (inhibition-off: eGFP and dsRed, inhibition-on: dsRed or luciferase as a readout). We have revised the text and figures to make this point clearer. Second, we are confident that our assays reflect *cis*-cleavage of the authentic viral polyprotein based on our cleavage site mutational analysis (**Figure 3**) and the fact that all *bona fide* / non-controversial inhibitors elicit clear dose responses in our gain-of-signal system (**Figures 4 and 5**). Moreover, the relative potencies of the inhibitors tested in our system reflect values reported in the literature.

In addition, the expression of the polyprotein may result in different levels of protein compared to what is present inside an infected cell. There is also likely difference in the localization and cellular compartmentalization of the polyprotein that takes place in an actual infected cell compared to the fragment of the protein expressed inside a reporter cell line. Therefore, it is possible that compounds that show activity in this assay may not be effective at blocking cis-cleavages in a cell infected with SARS-CoV-2.

Please see our response above.

Finally, the study seems premature and lacking in validation beyond a known inhibitor and a single protease target. There are no new screens performed and no new biology that results from this study. It therefore seems to lack the novelty, impact and new insight required for publication in Nature Communications.

To expand the relevance of our study and demonstrate the modularity of the system, we have

added 7 more proteases as described above (**Figure 4a-b**). We have also tested additional *bone fide* inhibitors as well as more controversial substances to help correct the literature (**Figure 4c**).

Please also note that this manuscript is under consideration for “*Communications Biology*” (not “*Nature Communications*”).

In addition, there are some minor issues that should be addressed in future versions of the manuscript.

Addressed as recommended below.

In figure 1 it is not clear what the right and left panels of cell images represent in each set of images. Are they the different time points before and after addition or removal of inhibitors? It does not seem that it is but it is not clear what is changing in the panels.

We apologize that these representative images caused confusion. We have removed them and instead use schematics to outline each system.

In figure 3 there should be labels for the X-axis for all graphs, not just the top graphs. The label for the bar graph X- axis is also missing in panel e.

We have corrected the axis labels as recommended.

REVIEWERS' COMMENTS:

Reviewer #1 (Remarks to the Author):

In their revised manuscript, the authors adequately addressed concerns identified in their original manuscript:

- 1) In the supplemental Figure 1 g-h, the signal stability of the stably transfected cells over time has been reported. While there is some fluctuation in the signal over time, the now added Z factor information indicates good assay performance over time.
- 2) The assay has been used to test the antiviral activity of eight proposed SARS-CoV-2 drugs, including the bona fide 3CLpro inhibitors PF-00835231 and PF-07321332. Especially the strong inhibitory activity of PF-07321332 validates this novel. However, information for PF-00835231 seems to be missing in Fig. 4c.

The manuscript has been further strengthened by adapting the assay to luciferase-based bioluminescence read-out, a sensitive and widely used platform used in many labs.

Reviewer #2 (Remarks to the Author):

The authors have addressed my concerns and have added new data showing the approach works for other targets. The manuscript is much stronger now and should be published.